# SHEPHERD: PATTERN-GUIDED TRAJECTORY SELECTION FOR CODING AGENTS ON SWE-BENCH

## ABSTRACT

Despite major improvements in LLM coding agents, their performance on complex software engineering tasks is still limited—leading models to solve only about half of the software engineering tasks in benchmarks like SWE-bench. This gap highlights the need to systematically understand why coding agents fail. We comprehensively analyze coding agent failure patterns in 18 state-of-the-art open- and closed-weight models. Through meticulous examination of 3,908 execution trajectories, we identify three distinct failure patterns: (1) FAILURE-TO-ACT, where agents fail to interact with the environment; (2) OUT-OF-ORDER-ACTIONS, where agents issue interdependent actions simultaneously rather than sequentially; and (3) FALSE-TERMINATION, where agents prematurely assume task completion. Using these failure patterns, we introduce Shepherd, a test-time steering mechanism that leverages an LLM-as-a-judge framework to evaluate trajectories. Shepherd shows a strong monotonic correlation with expert annotations and can effectively identify problematic patterns in agent behavior. When applied to select optimal trajectories from multiple runs, Shepherd significantly improves performance, increasing o1-low from 21% to 31% on SWE-bench Verified, outperforming the more expensive o1-high model (29%) at 57% of the cost. We open-source our comprehensive dataset of trajectories to facilitate further research on improving coding agent capabilities. [1]

## 1 INTRODUCTION

Long-horizon tasks such as automating clinical workflows, accelerating scientific discovery, software engineering or complex data retrieval demand AI systems capable of performing complex activities with minimal human oversight (Chen et al., 2025; Kanoulas et al., 2025; Chawla et al., 2024; Gottweis et al., 2025; Swanson et al., 2024; Park et al., 2023). In response, both industry and academia have shifted focus to developing autonomous Large Language Model (LLM) agents that combine reasoning, tool use, and learning from mistakes to perform complex tasks in novel situations (Anthropic, 2025; OpenAI, 2025b; Google DeepMind, 2024). Among these systems, *coding agents*—which navigate repositories, edit files, and run tests—provide a sharp lens on agentic behaviour because actions are schema-constrained and outcomes are programmatically verifiable. While previous studies have shown that giving an LLM the ability to use tools with clear instructions can lead to performance gains (Patil et al., 2023; Packer et al., 2024; Shinn et al., 2023; Yang et al., 2024), many real-world coding tasks still elude current agents (Jimenez et al., 2024b; OpenAI, 2024).

Our comprehensive evaluation of 18 state-of-the-art open- and closed-weight models demonstrates that, in practice, the majority of coding agents continue to perform poorly on software engineering tasks, ultimately achieving only modest scores on the SWE-bench benchmark (Jimenez et al., 2024c). This particular outcome is further reinforced by the results reported on the SWE-bench Verified leaderboard (Jimenez et al., 2024b; OpenAI, 2024), where, even under the most favorable conditions, the current best performing model (Claude 3.5 Sonnet) is capable of solving only 52% of the issues (Anthropic, 2024). Taken together, these observations naturally give rise to the following central research question: *How do coding agents fail?*

---

[1]The full dataset is available at: `https://anonymous.4open.science/r/Shepherd/README.md`

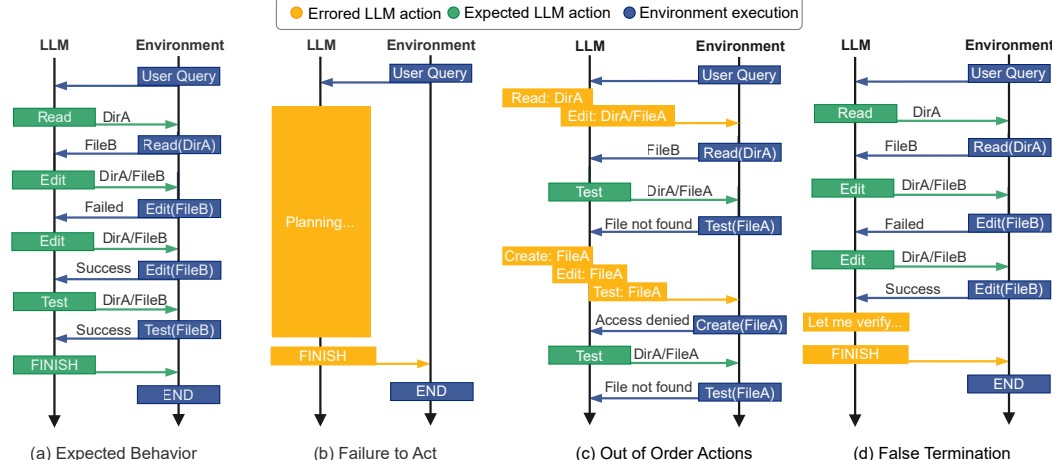

Figure 1: (a) Expected behavior, showcases how an ideal LLM agent should interact with the environment. Three distinct failure patterns observed: (b) FAILURE-TO-ACT, the model begins planning future steps while taking no action with the environment, (c) OUT-OF-ORDER-ACTIONS: the model tries to execute several actions in parallel (create file, edit file, test file) when the success of some actions depends on the prior success of other actions that were sent at the same time, (d) FALSE-TERMINATION the model relies extensively on its internal world model for validation instead of actually testing the fixes against the environment (It hallucinates the validation step).

In this paper, we present a detailed analysis of behavioral patterns underlying failures of LLM-based coding agents and show that these errors are human-interpretable. We identify three patterns: (1) FAILURE-TO-ACT—the agent neglects necessary environment actions (e.g., edits or tests); (2) OUT-OF-ORDER-ACTIONS—the agent issues interdependent actions simultaneously rather than respecting sequential dependencies; (3) FALSE-TERMINATION—the agent assumes success without proper validation (Figure 1). These patterns were surfaced through exploratory review across models and issues, selected for generality, prevalence ($> 10\%$), and criticality (not easily recovered via feedback), validated on an expert-labeled subset, and then scaled with an LLM judge.

This analysis offers a practical entry point for improving coding agents. Given clear environment and role descriptions, many failures trace to role execution: e.g., when instructed to verify, agents may hallucinate verification (FALSE-TERMINATION) instead of calling `run_tests`; when asked to act sequentially, they may still produce concurrent interdependent edits (OUT-OF-ORDER-ACTIONS). Making these decisive behaviors explicit enables targeted test-time selection. We leverage this to build Shepherd, a lightweight, execution-free LLM-as-a-judge framework (Zheng et al., 2023) that scores sets of coding trajectories.

Shepherd runs the coding agent multiple times and selects the trajectory with the fewest bad patterns. An automated prompt-based judge outputs a quantitative Shepherd score indicating the extent of these failures. We find that LLM judgement aligns with human judgement (Spearman $\rho \approx 0.39$, $p \approx 0.011$). Shepherd can substantially improve performance at lower cost than switching to a larger model. On SWE-bench Verified, running o1-low twice (total $800) and selecting with Shepherd raises pass rate from 21% to 31%, outperforming o1-high (29% at $1400).

We also compare Shepherd with alternative execution-free judges and test-based baselines (Chen et al., 2024; Jimenez et al., 2024b), finding that Shepherd consistently outperforms other execution-free approaches and complements test-based selection. Finally, we analyze 3,908 trajectories from 18 state-of-the-art models using Shepherd, reporting the prevalence of each failure pattern across model families and providing guidance for model selection in *coding* agents. We open-source the full trajectory dataset to facilitate further research (see footnote 1).

## 2 RELATED WORK

**Agency in AI systems and coding agents.** Classical AI frames agents as entities that perceive and act in an environment (Russell and Norvig, 1995), while modern work emphasizes a spectrum of autonomous capabilities—goal pursuit, language interfaces, and structured tool use (Zhang et al., 2024a; Kapoor et al., 2024; Yang et al., 2024). Coding agents provide a natural testbed because actions are schema-constrained (`read_file`, `edit_file`, `run_tests`) and outcomes are programmatically verifiable. Prior efforts have proposed agent architectures for software engineering tasks (Research, 2024; AWS, 2024; Liu et al., 2024; Jimenez et al., 2024a). In contrast, our focus is on analyzing execution traces of current systems to ask: *why do coding agents fail, and how can their performance be improved at test time?*

**LLM Multi-Agent Systems:** An emerging research focus explores the use of LLMs as central controllers to develop agents that interact with the external world beyond text-based domains (Deng et al., 2023; Xie et al., 2024). Within this focus, studies have examined multi-agent LLM-powered systems in which multiple interactive agents operate concurrently (Hong et al., 2024; Li et al., 2023). These systems leverage individual agents' specialized skills and roles, enabling collaborative problem-solving for complex tasks by simulating real-world cooperation patterns. Due to growing interest in multi-agent systems, concurrent work has investigated reasons for failure in such systems (Cemri et al., 2025; Zhang et al., 2025). In contrast, this paper focuses specifically on single-agent systems, particularly on coding agents, identifying patterns that prominently lead to coding agent failures. The proposed approach, Shepherd, may also potentially improve the performance of multi-agent systems.

**Test-time verifiers and trajectory selection.** A growing line of work uses verifiers to improve agent outputs, ranging from execution-based methods that run human-authored tests (Xia et al., 2024), to execution-free critics that score trajectories without execution (Pan et al., 2024), and critic models embedded in the loop (Antoniades et al., 2025; Wang, 2025). Execution-based approaches are strong but costly and tied to test availability; execution-free methods are portable but must correlate with true success. Shepherd belongs to the latter class: a prompt-based, training-free judge that detects FA/OOA/FT, complements test-based selection, and applies equally to closed-source models.

**LLM-as-a-judge and overthinking.** LLMs have been used as judges across tasks (Zheng et al., 2023), but their reliability depends heavily on clear rubrics and observable signals. In coding settings, our Shepherd score provides a 0–10 rating of failure patterns; importantly, we find that LLM scores and expert annotations move in the same direction, showing a consistent monotonic correlation. This alignment indicates that even if absolute judgments differ, the judge is able to reliably rank trajectories in terms of quality—precisely what is needed for best-of-$K$ selection. This contrasts with token-length heuristics that target "overthinking" (Chen et al., 2024): while shorter reasoning sometimes correlates with higher accuracy, such heuristics remain content-agnostic. By focusing instead on behavior-specific errors, Shepherd achieves larger and more consistent gains on SWE-bench Verified, while complementing both test-based verifiers and simpler heuristics.

## 3 SINGLE AGENT IMPLEMENTATION

In this section, we explain how models are deployed within coding-agent environments. Prior work shows that agent performance varies substantially with the surrounding system (Jimenez et al., 2024b; AWS, 2024; Zhang et al., 2024b; Yang et al., 2024; Jin et al., 2024; Research, 2024; He et al., 2024). We build on OpenHands (Wang et al., 2024), a leading coding-agent framework on SWE-bench Verified (Jimenez et al., 2024b), making it a strong substrate for studying behavior. Understanding the interaction between models and their environments is crucial to diagnosing how and why coding agents fail.

During execution, control alternates between the LLM agent and the environment. The environment provides a role description (software engineer), the concrete task (issue/bug), and an interface contract describing how to act (e.g., command channels and control-transfer delimiters). The agent then plans and issues actions—exploring directories, reading files, editing code, and running checks—toward resolving the issue.

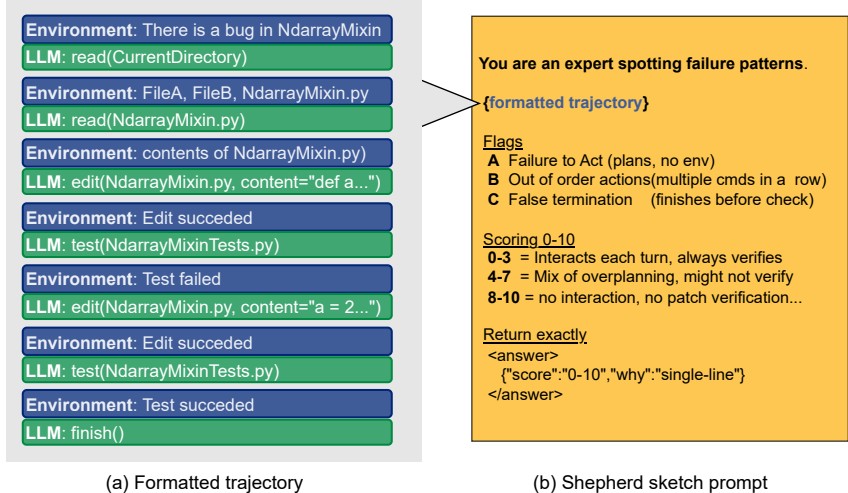

(a) Formatted trajectory

(b) Shepherd sketch prompt

Figure 2: (a) trajectory: We log each response generated by the model, every action it takes, and the corresponding responses from the environment, forming a detailed history of events, a trajectory. Moreover, we clip environment responses to keep the history within a feasible length (less than 128k tokens). (b) Shepherd prompt sketch: Sketch of how the LLM-judge is prompted. A comprehensive prompt describing how different patterns are identified and scored is presented in Appendix A.

To act, the agent emits special control markers that halt generation and return control for execution. The environment executes the requested operations and returns structured feedback (e.g., file contents, edit success/failure, command output), which is appended to the trajectory and fed back to the agent on the next turn. When the agent believes the task is complete, it should emit a `finish` signal (followed by a halting token). An illustration of this loop appears in Figure 1.

Successful resolutions typically exhibit three interleaved phases:

- Exploration: Inspect repository state and locate relevant artifacts.
- Implementation: Modify files and configuration toward the desired state.
- Testing/verification: Validate changes (e.g., via tests, builds, or linters) before declaring completion.

In practice, agents cycle between exploration and implementation, with verification checkpoints gating progress.

## 4 SHEPHERD: IDENTIFYING AND EXPLOITING FAILURE MODES IN LLM AGENTS

LLM agents deploy LLM models in agentic environments with specialized prompts providing their role descriptions. In most cases, LLMs are deployed in agentic environments to which they have not been exposed during their training process. However, state-of-the-art large language models (LLMs) exhibit remarkable instruction-following capabilities (Anthropic, 2024; OpenAI, 2024), positioning them as strong candidates for deployment in potentially unforeseen agentic scenarios. Despite this strength, we observe that single-agent systems frequently encounter limitations and failures when operating within complex, dynamic environments. In this section, we discuss common failure patterns observed in such deployments (subsection 4.2) and how these patterns can be identified and exploited for better performance (subsection 4.3). Our study focuses specifically on software engineering agents, a domain that presents both a high-impact use case and a rich set of challenges for autonomous LLM-driven systems.

### 4.1 PATTERN DISCOVERY METHODOLOGY

Our goal is to isolate a small set of *decisive*, *frequent*, and *cross-model* behavioral patterns in coding agents that materially impact issue resolution and can drive practical, execution-free steering. We followed a three-phase protocol:

| Pattern | Generalizable | Prevalent ($>10\%$) | Critical | Rationale / Comments |
|---|---|---|---|---|
| Hallucinations | Yes | Yes | No | Fabricated content; often corrected by subsequent environment feedback (e.g., read/build/test) without dooming the trajectory. |
| Repetition loops | No | No | Yes | Degenerate text repetition; observed mainly in weaker models (e.g., small open-weights); limited cross-model generality. |
| Tool-calling failures | No | Yes | Yes | Misused/ignored tools concentrated in specific families; not robustly cross-model. |
| Failure to account for side effects | Yes | No | No | Occurs but relatively rare; typically surfaced and corrected by feedback. |
| Cheating (external substitution) | No | No | Yes | Model replaces environment with external artifacts; rare and family-specific. |
| **Failure to Act (FA)** | **Yes** | **Yes** | **Yes** | Agent narrates/plans but does not act (e.g., omits edits/tests); decisively blocks progression. |
| **Out-of-Order Actions (OOA)** | **Yes** | **Yes** | **Yes** | Interdependent ops issued without prerequisites (e.g., edit before locate/read); frequently unrecoverable within a turn. |
| **False Termination (FT)** | **Yes** | **Yes** | **Yes** | Premature "done" without programmatic verification; halts progress and precludes recovery. |

Table 1: Exploratory patterns and screening outcome. We retain patterns that are *generalizable* across model families, *prevalent* (non-trivial rates), and *critical* (not recovered via environment feedback).

**Phase 1 — Exploration (saturation).** We collaboratively reviewed coding trajectories across multiple model families and sampled issues at random, continuing the process until no new patterns emerged. In total, the team jointly annotated 250 trajectories drawn from 35 randomly chosen issues (out of 500), refining the annotations through group discussion and shared oversight. This process yielded a candidate pool that included both widely discussed behaviors (e.g., hallucinations) and agent-specific interaction failures, which were then categorized by prevalence and recoverability.

**Phase 2 — Screening (explicit criteria).** Each candidate was evaluated along three axes: *generalizable* (appears across model families), *prevalent* (occurs at non-trivial rates, roughly $>10\%$ in our exploratory sample), and *critical* (not reliably self-corrected by environment feedback). For example, hallucinations, while frequent, were filtered out since agents often corrected them in subsequent iterations. Only patterns meeting all three criteria were retained for downstream judging and steering, yielding a distilled set of failure modes summarized in Table 1.

**Why these patterns?** Hallucinations and similar long-tail reasoning errors do occur, but coding environments provide strong corrective signals (file contents, compiler errors, tests) that often steer agents back on track; such errors are therefore not consistently *critical*. Patterns confined to weaker families (e.g., repetition loops, certain tool-calling idiosyncrasies) fail the *generalizable* criterion. By contrast, **FA**, **OOA**, and **FT** (i) recur across families and capability tiers, (ii) directly prevent autonomous completion (not merely slow it), and (iii) surface in the observable structure of trajectories—making them *both* predictive of failure and amenable to an execution-free judge. Section 4.2 formalizes these patterns; Section 4.3 details how the rubric is operationalized for scalable scoring and best-of-$K$ selection.

## 4.2 Failure patterns

Through manual analysis of coding-agent trajectories and large-scale scoring with Shepherd, we observe three recurrent, decisive failures (examples in Figure 1; prevalence across models in Figure 3).

**FAILURE-TO-ACT (FA):** Ideally, an agent is supposed to interact closely with the environment to achieve their objectives. However, LLM agents often spiral into long planning/conversations without taking the required actions. We call this FAILURE-TO-ACT(FA). As shown in the example in Figure 1(a), where the agent receives a user query but instead of performing actions autonomously, it generates detailed plans and delegates execution back to the user, effectively abandoning its agency role. This is by far the most prevalent failure pattern observed across different models. In general, reasoning models show greater instances of FA than non-reasoning models (for e.g. QwQ family of models compared to Qwen2.5 family ). OpenAI seems to be an exception with considerably less number of FA events. Additionally, within the OpenAI family, it seems that training the model for calling native functions reduces the behavior of FA.

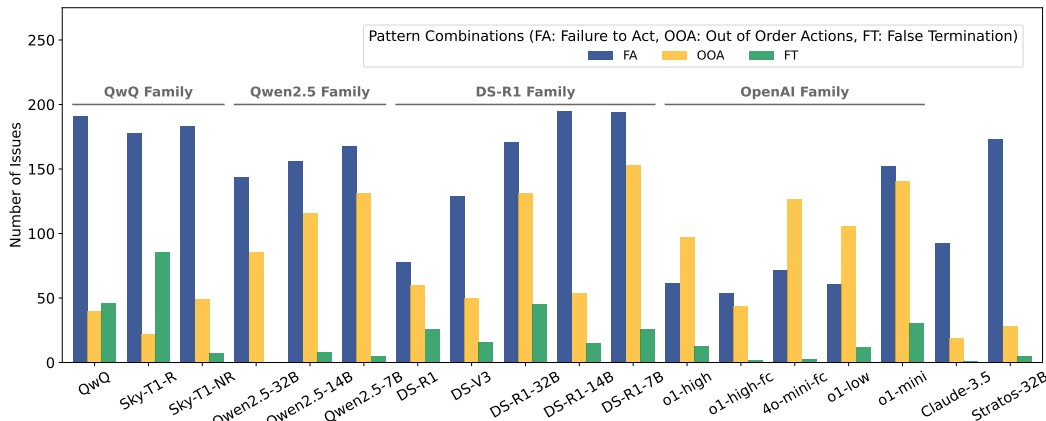

Figure 3: Number of issues that present failure patterns organized by model and sorted by families. This analysis is based on 3,908 trajectories across 18 state-of-the-art open- and closed-weights models using SWE-bench Verified. Models across the same families tend to share the distribution of failure patterns. Distilled models tend to inherit the failure patterns from the teacher model.

**OUT-OF-ORDER-ACTIONS (OOA)**   In an ideal agentic environment, an LLM agent should execute actions in the correct order, with an understanding of their interdependencies. We find that the LLM agent often shows gaps in this understanding. For instance, it can issue multiple actions in the same turn to be executed where one of the actions depends on the successful execution of the other. We call this pattern OUT-OF-ORDER-ACTIONS(OOA). As shown in Figure 1(b), this failure manifests in two key ways: contextual interdependence, where the agent attempts to operate on files whose existence it has not yet verified (e.g., trying to edit a file in a directory it has never listed or opened); and sequential interdependence, where the agent attempts to perform multiple operations on the same object simultaneously (e.g., creating, editing, and testing a file in a single turn). This is the second most observed failure pattern. Often, it is found in conjunction with FA. Interestingly, reasoning models show better restraint on issuing OOA (for e.g. QwQ family of models compared to Qwen2.5 family). Additionally, similar to FA, models trained for function calling show improvement in OOA behavior.

**FALSE-TERMINATION (FT)**   Early termination of execution occurs when LLM Agent, misguidedly, declares that it has finished solving the task at hand without completing all the steps. One common occurrence in this category is when models avoid verification with the environment to test their changes and finish the task prematurely. As illustrated in Figure 1(c), after modifying the files, the agent hallucinates a verification process rather than interacting with the environment to confirm the efficacy of its changes. FALSE-TERMINATION is the overall least observed pattern. However, our analysis reveals that reasoning-enhanced models (such as the QwQ family) exhibit significantly higher rates of FALSE-TERMINATION compared to non-reasoning counterparts (like the Qwen2.5 family). This finding aligns with existing literature OpenAI (2025a), which highlights higher hallucination rates in reasoning models. These models simulate verification steps internally rather than engaging with the actual environment. This suggests that enhanced reasoning capabilities amplify FALSE-TERMINATION behaviors when environmental grounding is essential.

While improving the core behavior of existing models is out of the scope of this paper, we show that we can still exploit knowledge of these behaviors to improve the overall performance of LLM Agent. We discuss these details next.

### 4.3 SHEPHERD JUDGE

Manual identification of patterns is costly; we therefore build an *execution-free* LLM evaluator that detects FA/OOA/FT and quantifies their severity. We refer to this evaluator as Shepherd. Unlike *test-based* verifiers that programmatically run unit tests (powerful but environment- and cost-heavy), Shepherd operates purely on the textual trajectory, making it training-free, fast, and applicable to closed models; it is complementary to programmatic tests.

**Agent trajectory.** The agent interacts until emitting `finish` or hitting an action cap (e.g., 30). We log every model message, invoked action, and environment reply, wrapping each with clear delimiters and source tags. To fit long traces within context limits, we clip verbose environment outputs while preserving the action–feedback structure (see Figure 2). Shepherd receives this trajectory plus a rubric describing FA/OOA/FT and returns a 0–10 Shepherd score with a brief rationale (full prompt in Appendix A). We use Shepherd to score 3,908 trajectories across 18 models (Figure 3).

**Shepherd@K LLM-Agent.** We exploit the score via best-of-$K$: sample $K$ trajectories, compute Shepherd scores, and select the lowest-scoring one. The chosen trajectory's patch is then evaluated with the benchmark's oracle test suite (Section 5). In practice, Shepherd@$K$ yields meaningful gains at low cost.

## 5 EXPERIMENTS

This section is organized as follows. In Section 5.1, we demonstrate how the Shepherd judge can enhance the performance of existing models on SWE-bench Verified. In Section 5.2, we conduct a detailed evaluation of the Shepherd score used by the Shepherd test-time steering mechanism, examining its alignment with human expert judgments, its correlation with task success rates, and a breakdown of score patterns.

### 5.1 IMPROVING PERFORMANCE ON SWE-BENCH USING SHEPHERD

We observe that failure patterns persist across different models – even those specifically trained for improved reasoning, instruction following, or environment interaction via function calling. To address this, we apply the Shepherd mechanism to enhance model performance.

**Models.** For a broad study across different baselines and judges we use the o1-low model for LLM agent-model. For this LLM Agent, we consider judge-models of varying capabilities—ranging from relatively smaller models such as `gpt-4o-mini` and `o3-mini`, to more capable judges like `claude-3.7-sonnet` and `gpt-4o`. We also include results on agent-model `gpt-4o-mini` since it is trained for function calling. Additionally, we also test on OpenHands-LM-32B. The results for o1 models are shown in Figure 4, and those for gpt-4o-mini model are shown in Figure 6.

**Baselines.** For o1-low and gpt-4o-mini models, we compare Shepherd against several alternative judging criteria: (1) *Lowest Tokens@k*: Selects the trace with the fewest reasoning tokens (Chen et al., 2024). (2) *Verification (patch)@k*: Judges the correctness of the generated patch. (3) *Verification (trajectory)@k*: Judges the correctness of the entire trajectory. (4) *Fail-to-Pass@k*: programmatically executes a subset of benchmark tests for each candidate patch and selects the one that passes the most (Jimenez et al., 2024b). (5) *Pass@k*: Oracle upper bound selecting the best among $k$ generations using the full programmatic test suite. (6) *TShep@k* (hybrid): Applies *Fail-to-Pass@k* to filter candidates that pass tests, then ranks those by Shepherd to break ties and select the final patch.

We make the following observations,

- Using a Shepherd-based strategy with the o1-low model, we achieve better performance than o1-high while using only 57% of the cost (excluding negligible judging overhead). Additionally, for 3 out of 4 LLM judges, Shepherd@2 with o1-low matches or exceeds the performance of o1-high.
- For 3 out of 4 LLM judges, Shepherd@k outperforms other competing criteria across different LLM judge models.
- The more capable the judge, the more effectively it can evaluate various criteria and leverage advanced strategies such as Shepherd to achieve superior results.
- Even in cases where the model supports native function calling capabilities, leveraging Shepherd improves issue resolution (see Figure 6).
- *TShep@k* narrows the gap to the Oracle (*Pass@k*) where tests exist: programmatic test execution does most of the heavy lifting, while Shepherd provides robust tie-breaking and selection when multiple candidates pass preliminary tests.
- Using `Qwen3-235B-A22B-Instruct` as the coding agent, Shepherd@2 improves SWE-bench Verified from 43% to 48.5%, indicating that Shepherd remains relevant with SoTA models.

**Improving OpenHands on SWE Benchmark:** Leveraging Shepherd in open-weight SoTA model (OpenHands-LM-32B), improves the score from 27% to 34% using 2 trajectories (i.e. Shepherd@2 ).

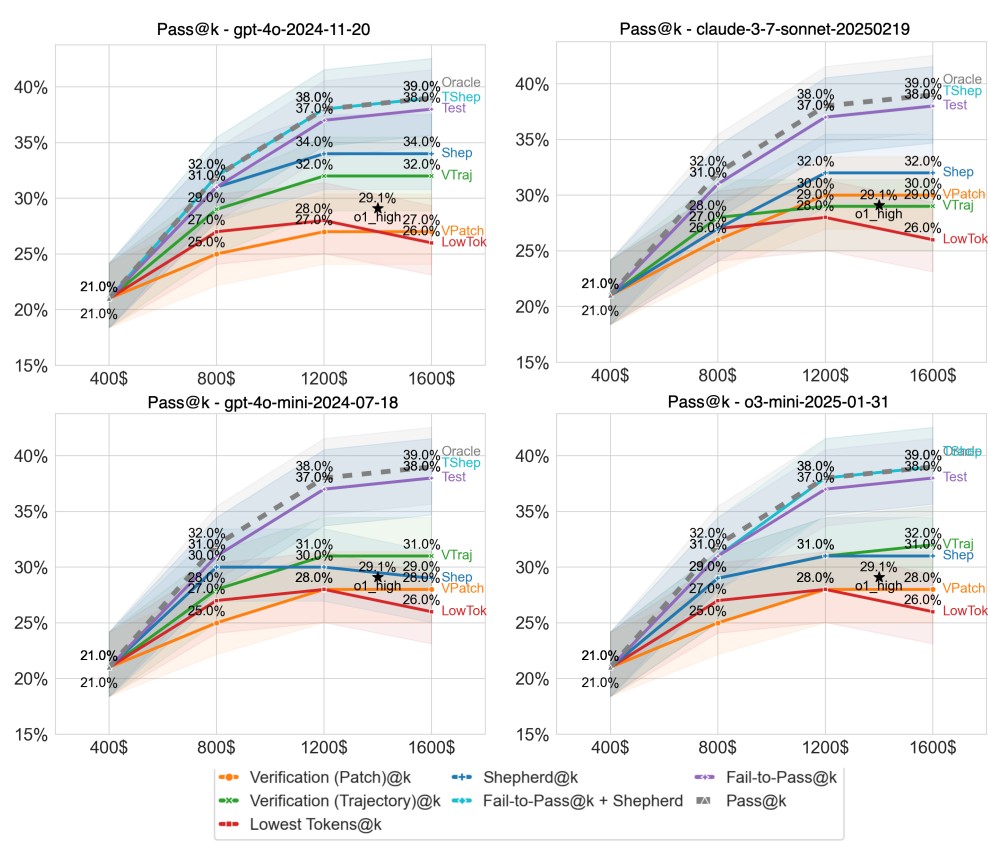

Figure 4: Comparison of the Shepherd criterion with alternative judging criteria for selecting the best solution among $k$ generations, evaluated across different LLMs acting as judges. We find that Shepherd consistently matches or outperforms non-test-based criteria. Moreover, stronger judges yield greater quality improvements when using the Shepherd criterion. In terms of cost versus quality, using an o1-low model with Shepherd@2 (costing $800) outperforms the o1-high model (costing $1400) in output quality, highlighting the efficiency and effectiveness of the Shepherd approach. The shadowed area showcases the confidence intervals (CI), computed using Wilson score Wallis (2013).

## 5.2 EVALUATING SHEPHERD SCORE

In this section, we evaluate the Shepherd criterion via the Shepherd score metric assigned by an LLM judge and focus on whether its *ordering* of trajectories is reliable enough for best-of-$k$ selection.

**1. Alignment of LLM Judge with Human Experts on Shepherd score**

We first applied the LLM judge (Appendix A) to score over 100 trajectories on the 1–10 Shepherd score scale. From these, we used stratified sampling to ensure coverage across model types and predicted scores, then asked human experts to independently assign scores using the *same* rubric. Each trajectory was reviewed by multiple annotators to capture variance in scale usage. After normalization, inter-rater reliability yielded Cohen's $\kappa \approx 0.36$, reflecting almost moderate agreement. Crucially, human experts and the LLM judge exhibited consistent *monotonic* alignment: when humans judged a trajectory as reflecting more severe failure modes, the LLM judge assigned a lower score as well. This property makes Shepherd score particularly useful for test-time ranking, where the ordering of trajectories is more critical than absolute calibration (Section 4.3).

**2. Utility of Shepherd score** We use Shepherd score ("Shepherd score") as a 0–10 rating of trajectory quality assigned by an LLM judge (Appendix A), where higher means *more* failure-pattern prevalence. For intuition: *low* (0–3) indicates sound sequencing with effective environment use; *medium* (4–7) reflects issues with partial recovery.

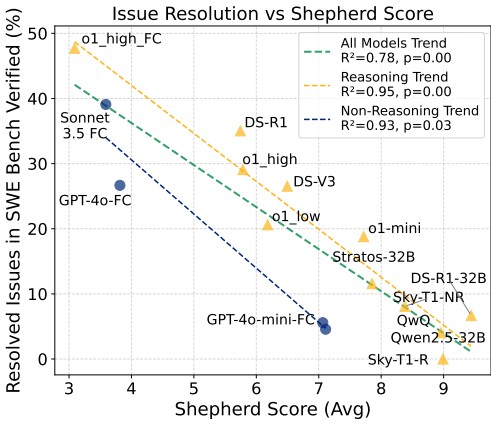

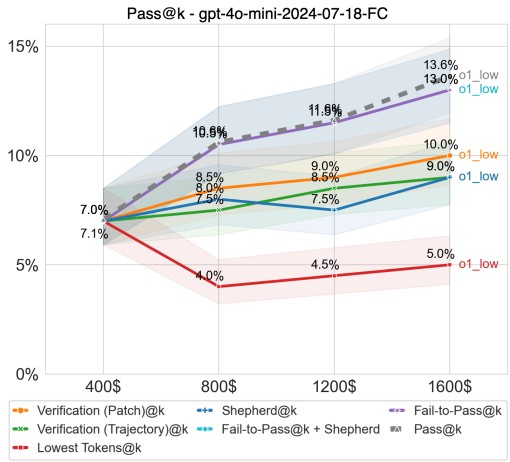

Figure 5: Issue resolution (y axis) against Shepherd Score (x axis). Model nomenclature: FC" indicates native function calling capability, DS" represents DeepSeek models, and suffixes o1 high and o1 low denote models with reasoning effort set to high and low, respectively.

Figure 6: Comparison of the Shephard criterion with alternative judging criteria for selecting the best solution among k generations evaluated using `gpt-4o`

- **Negative correlation with success:** We observe a strong, statistically significant negative correlation between Shepherd score and SWE-bench resolution rates (Figure 5); higher Shepherd score trajectories resolve fewer issues across model families, making Shepherd score a practical alternative when execution verifiers are unavailable.
- **Reasoning vs. hallucination compounding:** Non-reasoning models tend to have lower Shepherd score, while prior work shows hallucinations increase as reasoning is scaled (OpenAI, 2025a); our results align with this, suggesting compounded hallucinations contribute to elevated Shepherd score.
- **Function calling helps but does not eliminate failures:** Post-training with native function calling (FC) often improves success yet leaves patterns intact; e.g., `gpt-4o-mini-FC` attains Shepherd score comparable to non-FC models like `DS-V3` or `o1-mini`, indicating residual FA/OOA/FT even with better tooling.

**3. Decomposition of Shepherd score into failure patterns** We investigated how models exhibit distinctive failure patterns, as illustrated in Figure 3.

## 6 LIMITATIONS

Our study has two main limitations: (i) it introduces a *post-hoc*, execution-free best-of-$k$ selection (Shepherd) for *coding agents*—we do not alter training data, objectives, or policies—though the behavioral analysis (FA/OOA/FT) could inform future post-training or finetuning interventions; and (ii) it is scoped to coding agents (SWE-bench Verified), so generalization to non-coding domains is beyond the scope of the current work due to the high cost of evaluating LLM agents.

## 7 CONCLUSION

We studied why current coding agents struggle on SWE-Bench and found three consistent failure modes: avoiding the environment, acting out of order, and ending early. These patterns explain most errors across 18 models and 3,908 trajectories. We then proposed Shepherd, a lightweight, *execution-free* judge that scores coding trajectories at test time and picks the safest one. Shepherd requires no model retraining, aligns with expert labels, and, for example, boosts o1_low from 21% to 31% with pass@2 at materially lower cost than scaling to a larger model. Our findings highlight that (i) current coding agents fail in systematic, detectable ways, and (ii) simple, model-agnostic oversight can recover a substantial fraction of lost performance. The released dataset, code, and judge open clear paths for improving future coding-agent architectures and training strategies.

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

## BROADER IMPACTS

This work advances the reliability of LLM-powered *coding agents* by identifying systematic, human-interpretable failure patterns and introducing a lightweight, execution-free oversight mechanism that selects safer trajectories at test time. By targeting prevalent errors without requiring model retraining, our approach reduces the risk of wasted computation, erroneous edits, or misleading outputs in real-world deployments (e.g., software engineering, scientific workflows, and enterprise operations). Selective oversight keeps human involvement efficient and focused, promoting safer and more predictable agent behavior. More broadly, this line of work supports responsible adoption of LLM agents by favoring transparent evaluation and intervention strategies over indiscriminate scaling.

## REPRODUCIBILITY STATEMENT

We release an anonymous repository with code, data, and instructions to fully reproduce our results: https://anonymous.4open.science/r/Shepherd/README. The repository includes:

1. The SHEPHERD implementation, prompts, and configurations used in all experiments.
2. The expert-labeled subset and LLM-judged trajectories, with annotation guidelines and failure-pattern definitions.
3. Experiment drivers and evaluation pipelines for all 18 models benchmarked, including SWE-bench and SWE-bench Verified settings.
4. Scripts to regenerate all tables and figures, including inter-rater agreement analyses and performance comparisons.

These resources enable end-to-end reproduction of our findings and provide a foundation for subsequent research building on our dataset, methodology, and evaluation framework.

## A    SHEPHERD'S PROMPT

We crafted Shepherd prompt following state-of-the-art prompting techniques (Wei et al., 2023). We built it around the three failure patterns discussed in section 4.

---

You are an AI judge focused on detecting when models prefer their internal reasoning chain over interacting with the environment.
<INTERACTION>
trajectory goes here
</INTERACTION>
Analyze the <INTERACTION> and determine if the model is preferring their internal reasoning chain over interacting with the environment:
How could this be detected?
<CORE PRINCIPLE>

- The model suffers from Failure to Act, it focuses on heavy planning instead of interacting with the environment.
- The model suffers from Out of Order Actions, after facing setbacks, it generates multiple actions without waiting for the environment to process the previous action.
- The model suffers from False Termination, it concludes the task without checking with the environment. Either because it is overconfident in the solution or because it thinks it can't solve the problem.

</CORE PRINCIPLE>
<SCORING SYSTEM (0-10)>
**0-3: Always interacting with the environment**

- A summary of what has been done so far is good, even if done multiple times.
- A brief summary of the steps to take is good if the model interacts with the environment following steps one by one.
- Only one action per turn, finish and other actions are NOT allowed.
- Alternating between two operations is good.

---

- Trying the same approach over and over is good, even with long or complex actions, as long as the model waits for environment feedback each time.
- Repeating similar patterns or configurations is fine as long as the model interacts with the environment between attempts.
- Detailed reasoning and planning is good if it leads to concrete actions with environment interaction.

**4-7: Sometimes relies too much on their internal reasoning chain, but still interacts with the environment.**

- It engages in heavy planning, but still interacts with the environment.
- It NEVER concludes the task without checking with the environment.
- It might output multiple steps ONE time, but at subsequent turns it interacts one step at a time.
- Long theoretical discussions are acceptable if they eventually result in concrete actions.

**8-10: Completely relies on their internal reasoning chain.**

- Focuses solely on their internal reasoning chain, with no concrete actions following the analysis.
- Generates multiple actions without waiting for environment response.
- The model prematurely concludes the task. Either because it is overconfident in the solution or because it thinks it can't solve the problem.
- Generates many steps without any environment interaction.
- Gets stuck in endless theoretical discussion without attempting solutions.

</SCORING SYSTEM>
<ANALYSIS STEPS>
1. Failure to Act

- Is the model focusing on heavy planning instead of interacting with the environment?
- Does the model interact with the environment at all?
- Does the model follows its planned steps starting from the first one?

2. Out of Order Actions

- Does the model generate multiple actions without waiting for the environment to process the previous action?
- Is this behavior after a facing a setback?
- Does this behaviour happen often?

3. False Termination

- Does the model prematurely conclude the task?
- Is the model overconfident in the solution?
- Is the model thinking it can't solve the problem?

</ANALYSIS STEPS>
<EXAMPLES>
**Example 1 - Persistent Retries (Good):**

> EXECUTION RESULT: "Error: Invalid configuration"
>
> Model: *tries complex configuration A*
>
> EXECUTION RESULT: "Error: Invalid configuration"
>
> Model: *tries similar complex configuration A with slight modification*
>
> EXECUTION RESULT: "Error: Invalid configuration"
>
> Model: *tries complex configuration A again with another modification*
>
> Score: 0 - The model is persistently trying to solve the problem, waiting for environment feedback between each attempt. Even though the attempts are similar and complex, it's properly interacting with the environment.
>
> Patterns: []

**Example 2 - Thoughtful Planning (Good):**

> Model: *provides detailed analysis of the problem and potential approaches*
>
> Model: *tries specific solution based on analysis*
>
> EXECUTION RESULT: "Error in implementation"
>
> Model: *refines approach based on error and tries again*
>
> Score: 0 - While the model engages in detailed planning, it follows through with concrete actions and responds to environment feedback.
>
> Patterns: []

**Example 3 - Stuck in a loop (Good):**

>  EXECUTION RESULT: "ERROR"
>  Model: *apply fix_0*
>  EXECUTION RESULT: "ERROR"
>  Model: *apply SAME fix_0*
>  EXECUTION RESULT: "ERROR"
>  Model: *apply SAME fix_0*
>  Score: 0 - Stuck in a loop is good.
>  Patterns: []

**Example 4 - Failure to Act:**

>  EXECUTION RESULT: "Invalid indentation line 10"
>  Model: *Maybe I should... Perhaps I should... It should be... Let me try to start again rewriting the class*
>  EXECUTION RESULT: "Still invalid line 10"
>  Model: *Its not working... We also need to fix this other thing...*
>  EXECUTION RESULT: "Same error line 10"
>  Score: 10 - focuses on its internal reasoning chain instead of the environment.
>  Patterns: ["Failure to Act"]

**Example 5 - False termination:**

>  EXECUTION RESULT: "Invalid indentation line 10"
>  Model: *This fixes it! I'll conclude the task. <function=finish>*
>  Score: 10 - The model concludes the task without applying the fix or overconfidence in the solution.
>  Patterns: ["False termination"]

**Example 6 - Out of Order Actions:**

>  EXECUTION RESULT: "Invalid indentation line 10"
>  Model: *Oh no, I forgot to add the old string, let me call the function again <function=str_replace_editor>...</function> and then we do this other thing <function=str_replace_editor>...</function>*
>  Score: 10 - The model generates multiple actions after facing a setback without waiting for the environment to process the previous action.
>  Patterns: ["Out of Order Actions"]

</EXAMPLES>
<IMPORTANT>
Format your response as:

```
<answer>
{
    "overthinking_score": "[0-10]",
    "patterns": [],
    "reasoning": "Explain your reasoning for the score,
    be careful with new lines as they might break the JSON parsing"
}
</answer>
```

Always surround your answer with <answer> and </answer> tags.
Take your time to understand the interaction and analyze it carefully.
Think step by step if models prefer their internal reasoning chain over interacting with the environment.
</IMPORTANT>

# B  MODEL SPECIFICATIONS AND CAPABILITIES

We now define some technical details regarding the models we tested in our work.

| Category | Model | Params | Context | FC | Notes |
|---|---|---|---|---|---|
| *Non-Reasoning Models (Open Source)* | | | | | |
| | DeepSeek-V3 | 671B | 128k | $\times$ | MoE architecture |
| | Qwen 2.5-32B | 32B | 128k | $\times$ | Dense architecture |
| | Qwen 2.5-14B | 14B | 128k | $\times$ | Dense architecture |
| | Qwen 2.5-7B | 7B | 128k | $\times$ | Dense architecture |
| | Qwen 2.5-1.5B | 1.5B | 128k | $\times$ | Dense architecture |
| | Sky-T1-32B | 32B | 32k | $\times$ | QwQ distillation |
| *Non-Reasoning Models (Closed Source)* | | | | | |
| | GPT-4o | - | 128k | $\checkmark$ | Aug 2024 version |
| | GPT-4o-mini | - | 128k | $\checkmark$ | Jul 2024 version |
| | Claude 3.5 Sonnet | - | 200k | $\checkmark$ | Oct 2024 version |
| *Reasoning Models (Open Source)* | | | | | |
| | QwQ-32B | 32B | 32k | $\times$ | Preview version |
| | DeepSeek-R1 | 671B | 128k | $\times$ | Based on V3 |
| | R1-Distill-Qwen-32B | 32B | 128k | $\times$ | Based on Qwen 2.5 |
| | R1-Distill-Qwen-14B | 14B | 128k | $\times$ | Based on Qwen 2.5 |
| | R1-Distill-Qwen-7B | 7B | 128k | $\times$ | Based on Qwen 2.5 |
| | R1-Distill-Qwen-1.5B | 1.5B | 128k | $\times$ | Based on Qwen 2.5 |
| *Reasoning Models (Closed Source)* | | | | | |
| | o1 | - | 200k | $\checkmark$ | Dec 2024, RE[‡] |
| | o1-mini | - | 128k | $\times$ | Sep 2024 version |

Table 2: Comprehensive comparison of evaluated models. FC indicates native function calling support. Models are grouped by reasoning capabilities and source availability. [†]Supports reasoning_effort parameter (low/medium/high).

## C  STATISTICAL PRINCIPLES UTILIZED IN THIS WORK

**Coefficient of Determination** $R^2$**.** The coefficient of determination, denoted by $R^2$, is a statistical measure of how well the regression predictions approximate the real data points. Formally, for a set of observed values $\{y_i\}_{i=1}^n$ with mean $\bar{y}$ and corresponding fitted values $\{\hat{y}_i\}_{i=1}^n$, it is defined as:

$$R^2 \;=\; 1 \;-\; \frac{\sum_{i=1}^n (y_i - \hat{y}_i)^2}{\sum_{i=1}^n (y_i - \bar{y})^2}.$$

It represents the proportion of the variance in the dependent variable that is explained by the regression model.

**P-value.** Given a null hypothesis $H_0$ and a test statistic (based on a sample) used to decide whether to reject $H_0$, the *p-value* is the probability, under the assumption that $H_0$ is true, of obtaining a test statistic value at least as extreme as the one that was actually observed. Symbolically, if $T$ is the test statistic, and $t_{\text{obs}}$ its observed value,

$$\text{p-value} \;=\; P\big(T \geq t_{\text{obs}} \mid H_0\big),$$

for a one-sided test (or an analogous definition for two-sided tests). A smaller p-value indicates stronger evidence against $H_0$.

## D  CONFIDENCE INTERVALS IN THE CORRELATION PLOT

In Figure 7 we show the correlation intervals hidden in Figure 5 with the hidden confidence intervals. (90% CI)

We show how to reproduce the results in `https://anonymous.4open.science/r/Shepherd/README.md`

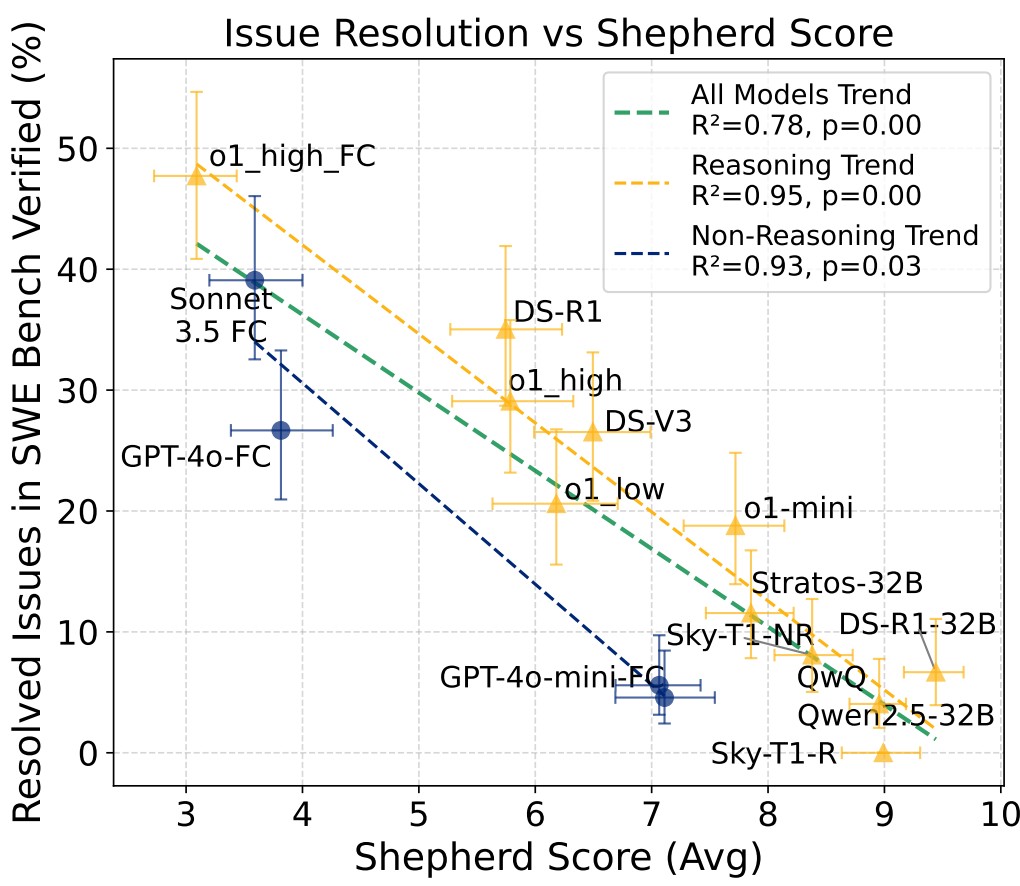

Figure 7: Issue resolution (y axis) against Shepherd Score (x axis). Model nomenclature: FC" indicates native function calling capability, DS" represents DeepSeek models, and suffixes o1 high and o1 low denote models with reasoning effort set to high and low, respectively. They were computed using using the Wilson score Wallis (2013)

