# OpenReview forum: "Shepherd: Pattern-Guided Trajectory Selection for Coding Agents on SWE-Bench"
_ICLR.cc/2026/Conference — Submitted to ICLR 2026_

### Official Review · Reviewer_Rd19 · 2025-10-18

**Soundness:** 3
**Presentation:** 3
**Contribution:** 2
**Rating:** 4
**Confidence:** 4

**Summary:**

This paper analyzes the common failure modes of coding agents on the SWE-bench benchmark. The authors propose a taxonomy of three human-interpretable failure patterns: Failure-to-Act (FA), Out-of-Order-Actions (OOA), and False-Termination (FT) studying  3,908 trajectories across 18 models. Based on this taxonomy, the paper introduces Shepherd, a test-time, execution-free LLM-as-a-judge that scores agent trajectories. The core method, Shepherd@K, selects the best trajectory from K samples based on the lowest failure score. The paper shows that Shepherd@K (best-of-K by lowest failure score) improves the final success rates. The authors report alignment between Shepherd scores and human judgments and release code/data for reproduction.

From reviewer's point of view, the main contribution of the paper is a set of interpretable failure modes that can be used further to improve agents by training. The novelty and performance gains are weak-to-moderate and comparable to more simple baselines.

**Strengths:**

1. The analysis of failure patterns provides valuable insights, contributing to a taxonomy of errors and promising research directions for improving coding agents. FA/OOA/FT are concrete, observable in traces, and argued to be general, prevalent, and critical; the paper motivates why other candidates were excluded.

2. The clear and interpretable rubric used by the LLM-as-a-judge enhances transparency. This rubric could be adapted for future work, such as rubric-based model training together with RLVR.

3. The failure patterns are well-illustrated, and the paper is generally well-written and accessible.

**Weaknesses:**

1. To ensure generalizability and address potential selection bias, the evaluation could be strengthened by selecting pattern failures from a held-out set of tasks outside the SWE-bench Verified dataset. This would help confirm that the identified failure patterns are not specific to the benchmark.

2. Shepherd demonstrates modest performance gain, which is often comparable to the Verification(trajectory) baseline (Figure 4, 6). It is also likely that a simple, trained Outcome Reward Model (ORM) that learns to differentiate between good and bad trajectories would outperform this heuristic-based approach. Though it's clear that Shepherd provides additional interpretability features.

3. The prompt engineering in Appendix A contains a potentially counter-intuitive example. Example 3, "Stuck in a loop (Good)," assigns a perfect score of 0 to an agent that repeatedly applies the same failing fix. This seems to reward unproductive behavior, which warrants clarification.

4. The reported Spearman correlation between the LLM judge and human experts (ρ≈0.39) is weak to moderate, which raises questions about the reliability and robustness of the LLM-as-a-judge.

5. (minor) The paper mentions a three-phase pattern discovery protocol, but only two phases (Exploration and Screening) are explicitly detailed. The third phase seems to be the final selection of patterns based on the screening criteria, which could be stated more clearly.

**Questions:**

1. Could you clarify the distinction between the Pass@k and Fail-to-Pass@k baselines? Fail-to-Pass tests are typically the oracle verifiers for a given task. Does Fail-to-Pass@k use only a subset of the full test suite that Pass@k uses? If so, should this baseline still be considered a form of oracle verification?

2. There is a notable discrepancy between the performance of DS-V3 reported in this paper (~27% in Figure 5) and results from other sources (e.g., the OpenHands leaderboard, which claims scores 32.4% for Deepseek-V3-1226 and 38.8% for V3-0324 ). How do you account for this performance gap?

3. Regarding the human annotation agreement (κ \approx 0.36), could you elaborate on the primary sources of disagreement?

4. Could you provide an error analysis for cases where Shepherd fails? Specifically, what are the characteristics of trajectories that Shepherd scores favorably but ultimately fail the task, and vice-versa?

5. How does performance scale with K in Shepherd@K? Is there a saturation point where increasing K yields diminishing returns?

6. There appears to be a contradiction in Section 5.2. The text states: "when humans judged a trajectory as reflecting more severe failure modes, the LLM judge assigned a lower score as well". However, the Shepherd rubric is designed so that higher scores indicate more severe failures. Please clarify if this is a typo.

---

### Official Review · Reviewer_wMiR · 2025-10-31

**Soundness:** 2
**Presentation:** 3
**Contribution:** 2
**Rating:** 4
**Confidence:** 5

**Summary:**

This paper proposes a test-time approach to improve coding agents by sampling multiple trajectories and selecting the most promising one using Shepherd, an execution-free LLM-as-a-judge framework. Through manual inspection of trajectories the authors identify three main failure patterns that they claim are generalizable, prevalent, and critical across both closed and open source model families:
- Failure-to-act: the agent stops interacting with the environment
- Out-of-order actions: the agent does not follow the right order when executing actions
- False termination: the agent declares success without performing proper verification

Based on these findings, the authors build a prompt that can be used by Shepherd to select the most promising trajectories. This presented method allows for relevant improvement (o1-low improves from 21% to 31% on SWE Bench Verified) without changes to the agent itself.

**Strengths:**

- The paper is easy to read and well presented: the idea is simple and easy to implement. The dataset and code are also made available.
- Shepherd shows nice improvement for smaller models (o1-low beats o1-high on SWE Bench Verified, OpenHandsLM-32B improves as well), which is a meaningful result from a budget and resource perspective.
- The failure taxonomy is clear, novel, and validated across multiple models (Figure 3), providing also useful insight into how training choices (reasoning vs non-reasoning) impact agent behavior.
- The manual analysis is extensive and leads to a small, actionable set of failure modes that are easy to incorporate into the LLM judge, without requiring extra training.

**Weaknesses:**

1. The paper relies on manual inspection to identify recurring failure modes but does not propose or evaluate automated methods to discover new patterns. I believe this limits the generalizability and applicability of this work, as new model are released quickly and these patterns might not appear in the next generation of models.
2. The paper states that coding agents are still performing poorly on coding tasks, such as SWE Bench Verified. However, this sounds a bit outdated at the time of the submission – SWE Bench Verified has now models reaching almost 70%.
3. Limited novelty and lack of detailed analysis: inference-time selection and LLM-as-a-judge are well studied, also in the context of SWE Bench Verified (two relevant works that are not cited are "CodeMonkeys: Scaling Test-Time Compute for Software Engineering" and "R2E-Gym: Procedural Environments and Hybrid Verifiers for Scaling Open-Weights SWE Agents"). While the analysis of failure patterns is useful, the analysis is not thorough enough for a venue like ICLR:
    - Insufficient analysis of complementarity with test-based methods: TShep seems to leverage mostly test-based selection rather than Shepherd ranking.
    - The paper assumes agents will be automatically able to solve some of the issues by simply sampling multiple times: it would be interesting to see if the insights from the three patterns could be used to guide the agent during inference (“serial” test time compute instead of looking only once trajectories are generated).
    - No insights on the newly resolved issues
    - Shepherd is tested only at selecting between a few trajectories (k=2): would Shepherd be robust when scoring multiple trajectories? Inference scaling might add an additional increase with k=8 or k=16.
4. Baselines need more details: Verification(patch)@k and Verification(trajectory)@k are not described clearly enough to understand the comparison, while for Fail-to-Pass@k, it is not clear which tests are used and if they would be available in a real-world scenario.

**Questions:**

- Question related to Weakness #3: How does each pattern contribute to the increased number of resolved issues?
- Question related to Weakness #4: Which kind of tests are used for the Fail-to-pass@k method? Are these the same fail-to-pass tests used to evaluate final success? If so, these tests may not be available in many real-world settings, leaving Shepherd@k as the only option, which has much lower performance.
- The paper states that Shepherd provides useful tie-breaking (line 373) when test signals are insufficient: however, in Figure 4 I don’t clearly see an instance where TShep strictly improves upon plain test filtering — can you clarify and provide quantitative evidence of cases where Shepherd provides value after test filtering?

Minor Comments:
- Figure 6 places o1_low on the far right, inconsistently from Figure 4.

---

### Official Review · Reviewer_X6yH · 2025-10-31

**Soundness:** 3
**Presentation:** 4
**Contribution:** 3
**Rating:** 4
**Confidence:** 5

**Summary:**

The author introduce Shepherd, as post-hoc coding-agent trajectory steering approach which uses a rubric to score trajectories. The rubric was developed by manually investigating failure modes of OpenHands trajectories. The authors analyzed 8 failure patterns of which 3 seem to generalize, are prevalent and critical. These patterns are 1) Failure to act (agent not using environment), 2) Out-of-order actions, and 3) False-termination.  On the experimental side, Shepherd is evaluated against a solid set of baseline post-hoc selection methods. It shows small absolute improvement in term of average pass rates.

**Strengths:**

* The authors perform manual analysis of trajectories rather than relying on LLM interpretations
* Baselines are chosen thoroughly (oftentimes this is overlooked)
* Presentation of the work is really good and it is easy to follow
* Performance of correlation analysis with human judgement
* Visualization of CIs to understand performance more holistically
* Figure 5 shows strong correlation of Shepherd score with resolve rate
* Making trajectories publicly available

**Weaknesses:**

* Only OpenHands trajectories are evaluated which makes it hard to show generalizability of the method.
* Trajectories are all based on python repositories. It is unclear if agents run into the same failure patterns for other languages
* Correlation of Shepherd score with human annotations is weak (not really a weakness given that we don't know how a human post-hoc patch selector performs)
* Some of the identified patterns may be a artifacts from the agent scaffold not using tools properly (e.g., sequential interdependence). This may be reduced by using stronger models like Haiku/Sonnet 4.5.
* Experimental results show that Shepherds performance lies largely in the CIs of competing methods, rendering improvements largely not statistically significant.
* We still rely on strong LLM to be able to judge the rubric well
* Throughout the text you highlight alignment with human labels but the correlation analysis shows only weak correlation.

**Questions:**

* Could you evaluate Shepherd on at least one other scaffold such as SWE-Agent or Aider? I am willing to raise my score if you can show that Shepherd generalizes over at least one other coding agent and one other programming language.
* Since you have human annotations for some trajectories, would it be possible to add them as a baseline? This could have quite some impact as the community is unsure about whether a correlation between LLM and human annotation results in better outcomes.
* In 5.2 you analyze inter-rater reliability. Is that between LLM and human annotations? If so, what's the expected inter-rater reliability of humans alone and how does adding an LLM annotator change that score?
* Line 306 1(c) -> 1(d)

---

### Official Review · Reviewer_PYgk · 2025-11-03

**Soundness:** 2
**Presentation:** 3
**Contribution:** 2
**Rating:** 2
**Confidence:** 5

**Summary:**

This paper analyzes coding agent failure patterns in coding agent trajectories and identifies three distinct failure patterns:  (1) FAILURE-TO-ACT, where agents fail to interact with the environment; (2) OUT-OF-ORDER-ACTIONS, where agents issue interdependent actions simultaneously rather than sequentially; and (3) FALSE-TERMINATION, where agents prematurely assume task completion. Then, the authors use these patterns to design a LLM-as-a-judge prompt system called Shepherd that can be used to evaluate trajectories and potentially correct the agent course of action.

**Strengths:**

**Ease of Adoption**: Since Shepherd is a prompt-based approach is very easy to adopt and implement in the existing agentic systems.

**Experiment Design**: The paper covers many alternative judging criteria as well as models which makes the experiment results to be insightful.

**Weaknesses:**

**Connection with Prior work**: There are other RW that look into agent failure modes. For instance, the famous swe-agent paper identifies 8 categories of failure modes. Similarly https://arxiv.org/pdf/2503.07832 identifies three failure modes in refactoring tasks. Although the authors have cited the swe-agent paper for the agent work, they have not cited their categorization of failure modes and how those categorizations relate to the proposed categorization in the paper.

**Lack of Application**: The failure mode categorizations in the paper are very high-level and while being high-level helps and LLM-as-a-judge system like Shepherd to identify them, it doesn't give agent developers enough information to be able to address these failure modes and therefore is not very applicable in practice.

**Impact**: Figure 4 shows that the execution based approaches are superior than Shepherd. Given that the execution-based appraoches can also be integrated in the model response to get better results, it's not clear what the real impact of Shepherd is. At least for swe-bench were test execution is not an issue.

**Questions:**

- The authors analyzed openhands trajectories on swe-bench. Did they use a test/evaluation specific prompt in open-hands? They report "failure to act" as the main reason for failure. However, for safety reasons many coding agents don't proceed with their plans unless it's verified by the user. Therefore, evaluations should perform on specific test/evaluation prompt which tells the agent that it's job is to only evaluate the system and it should not verify the plan with the user. Getting trajectories with normal prompt will result in "failure to act" category by design because of safety reasons.

---

### Meta-Review · Area_Chair_uEWT · 2026-01-17

**Summary:**

All of the reviewers recommended reject initially. The main concerns were limited novelty and relation to prior work, the generalizability of the findings, weak correlation with human judgement, and issues with the experimental design and baselines.

**Reviewer Concerns:**

There was no rebuttal.

**Reviewer Scores:**

There was no rebuttal and the reviewers all recommend reject.

---

### Decision · Program_Chairs · 2026-01-26

Reject